# Uncertainty-aware predictive modeling for fair data-driven decisions

**Patrick Kaiser**
School of Social Sciences
University of Mannheim

**Christoph Kern**
Department of Statistics
LMU Munich

**David Rügamer**
Department of Statistics
LMU Munich

## Abstract

Both industry and academia have made considerable progress in developing trustworthy and responsible machine learning (ML) systems. While critical concepts like fairness and explainability are often addressed, the safety of systems is typically not sufficiently taken into account. By viewing data-driven decision systems as socio-technical systems, we draw on the uncertainty in ML literature to show how fairML systems can also be safeML systems. We posit that a fair model needs to be an uncertainty-aware model, e.g. by drawing on distributional regression. For fair decisions, we argue that a safe fail option should be used for individuals with uncertain categorization. We introduce semi-structured deep distributional regression as a modeling framework which addresses multiple concerns brought against standard ML models and show its use in a real-world example of algorithmic profiling of job seekers.

## 1 Introduction

Technological developments have led to an automation of decision making processes in various contexts, including employee recruitment [34], personalized medicine [45], jurisdiction [24], finance [17], and disaster management [44]. Automated decision-making (ADM)[1] systems aim to advance timeliness, efficiency, quality, and transparency in the allocation of resources and interventions [36]. While research shows that this can lead to many positive developments [21], new issues arise as well. The Panel for the Future of Science and Technology of the European Parliament, for example, highlights that a "good" (data-driven) decision system needs to take ethical, political, legal, and technical issues into account [8]. As such, fairness, transparency, explainability, and accountability concerns need to be addressed next to algorithmic performance.

An additional aspect, which so far has not been discussed as much in this context, is safety. Any technology which is to be implemented safely in a socio-technical system such as ADM needs to be aware of its uncertainty [43]. While research on uncertainty in machine learning (ML) is progressing [20], less attention has been paid to how uncertainty-aware prediction modeling can cater toward a safe and fair deployment of ML in data-driven decision-making systems.

[5] highlight the importance of uncertainty for socially responsible decision systems. As fairness is understood in terms of social bias, consequences of uncertainty for measurement bias and representation bias are discussed. Measurement bias can lead to serious issues for existing bias mitigation methods,

---

[1]We refer to ADM system as any not solely human-based decision system, while (non-)data-driven decision system specifies explicitly if decisions are based on inference from data or not.

2022 Trustworthy and Socially Responsible Machine Learning (TSRML 2022) co-located with NeurIPS 2022.

while representation bias urges for collecting more data for certain groups. As representation bias (alias differences in epistemic uncertainty) is sufficiently presented, we want to focus on the topic of measurement bias (alias differences in aleatoric uncertainty), and particularly on methodological approaches for uncertainty quantification for social groups. When it comes to decision making, [5] highlight the "reject option" as a way to combine human and data-driven decision making. On this basis, we focus on the implementation and consequences of the "reject option" for ADM systems.

**Our contribution**    In this paper, we connect fairness and uncertainty considerations in the context of data-driven decision-making. We argue that in order to advance towards fair ADM, uncertainty needs to be taken into account both at the prediction step (building a prediction model) and the decision step (acting based on predictions) of ADM systems. At the prediction step, fair models need to be aware and transparent about uncertainty: Not only the predictions, but also the uncertainty can vary across social groups as the training data may be differentially informative for different subpopulations. At the decision step, individuals should not be fitted into an actionable category if the uncertainty is too high to justify an action. We therefore put forward and implement the concept of a safe fail option in ADM to progress toward fair data-driven decision-making under uncertainty.

We introduce how distributional regression, and in particular semi-structured deep distributional regression (SSDDR, Rügamer et al. 38), can be used to acknowledge uncertainty at the prediction and decision step of ADM systems. As SSDDR can be used to build performant models that are also interpretable with respect to sensitive features, it allows to meet multiple concerns that have been raised in ADM contexts within the same modeling framework. We demonstrate the use of SSDDR for fair and uncertainty-aware ADM with an empirical example of algorithmic profiling of job seekers.

## 2    Data-driven decision systems

### 2.1    Background and setup

Data-driven decision systems can be framed as socio-technical systems [12], which use data either to automate or to assist a decision process [36]. A prominent example are public profiling systems [11]: Decisions are made about the allocation of public resources, while resources are scarce [28]. Profiling of the unemployed, for example, aims to efficiently allocate programs to job seekers in order to maximize their reintegration chances into the job market [25, 28]. In this context, data-driven decision systems are typically semi-automated systems as they are supposed to support (not replace) caseworkers in selecting individuals who are eligible to participate in support programs [23]. In order to assess such systems, humans, technology and their interplay need to be jointly investigated [12].

From a decision theoretic perspective, a data-driven decision system includes a **prediction task** and a **decision task** [26]. The prediction task contains everything about handling the data, from data generation to data modeling. The data used in this step can include sensitive attributes (such as gender, ethnicity, or religion) that may be protected by anti-discrimination law (for the U.S., see [2, 30]). The prediction model outputs one or multiple numbers (scores), which are then used as input in the decision task and turned into a decision. In non-data driven decision-systems, on the other hand, there is no prediction task: the decision task is solved solely by human experts or by predefined rules.

**Prediction task**    Given model $f : \mathcal{X} \times \mathcal{A} \to \mathcal{Y}$ trained on data containing fairness relevant attributes $\mathcal{A}$ and other features $\mathcal{X}$, what is the prediction $\hat{Y}$ of an outcome $Y \in \mathcal{Y}$ for a new individual $(X, A) \in \mathcal{X} \times \mathcal{A}$?

**Decision task**    Given some information $\hat{Y}$ about the new individual $(X, A)$, which discrete (and often binary) decision $d$ should the system output?

### 2.2    Fairness

Traditionally, fairness in ML is often framed in terms of (social) bias in order to find technical solutions to mitigate bias and achieve fairness [4, 9, 30, 32]. However, as data-driven decision systems are usually socio-technical systems, fairness needs to be seen from a broader perspective [7]. One way of connecting the technical and social perspectives is by mapping fairness considerations to the data-driven **prediction task** and the not necessarily (only) data-driven **decision task** [26].

**Prediction task**   Fairness in the prediction task requires fairness on the level of the model: Traditionally in ML, a loss function is chosen using considerations about the data generating process. It is optimized equally for all entities in the data [18]. Both the variables to be included and the model to be used are chosen optimally given the loss. From a fair machine learning (fairML) perspective, however, considerations about the social realities of the individuals have to be included in those modeling choices [15]: Which variables are justified to influence the outcome? Which loss best reflects social realities? How are the losses of the individuals weighted?

**Decision task**   Fairness in the decision task questions (distributive) justice: Traditionally in ML, decisions are computed directly by thresholding the predicted score of the model, usually using the predicted probability for a class as the score in a classification setting [7]. However, considerations about distributive justice principles are necessary to construct a system that aims to be fair [26]: When and how is it justifiable to treat individuals differently? When and how is it justifiable to treat social groups differently?

## 2.3   Uncertainty

In the ML literature, uncertainty is generally divided into **aleatoric** and **epistemic** uncertainty [20]. Aleatoric uncertainty deals with randomness inherent in the data generating process. Epistemic uncertainty deals with the uncertainty that is reducible when increasing the data size. Hence, the more data available, the better the model can reflect the true underlying process. In the ML process, uncertainty can be considered by probabilistic methods, set-based methods, and combinations. [20].

Uncertainty in a predictive system can be acknowledged by outputting not only one statistic in the prediction of $\hat{y}$, but a set of predictions, a fully specified distribution of $y$ or a set of distributions [20]. In the regression setting, aleatoric uncertainty is expressed by the conditional probability distribution of $Y|(X,A)$ or by probabilistic sets of $Y|(X,A)$. Distributional regression is the concept of modeling all parameters of a parametric distribution of $Y|(X,A)$ [37]. Therefore, the uncertainty quantification is valid only given the assumed parametric assumption [10]. However, the parametrization in distributional regression allows direct interpretability [42]. Quantile regression [31] and conformal predictions [41] create probabilistic sets without distributional assumptions, but lack the possibility of direct interpretability. Quantile regression and distributional regression directly change the model loss, while conformal predictions can be seen as a post-hoc technique to quantify uncertainty for any black-box models. Combinations of both approaches have been shown to create the best uncertainty quantification [10]. Epistemic uncertainty, however, needs to be considered in other ways, e.g. by dropout variational inference in Bayesian neural networks [14].

In the classification setting, the probabilistic predictions of a given class already denote the aleatoric uncertainty. Quantification of aleatoric uncertainty therefore translates into the issue of model calibration [20]. Hence, when the probability for all $c$ classes equals $\frac{1}{c}$, aleatoric uncertainty is maximal, as we do not know more than random guessing [27]. Epistemic uncertainty can be modeled by predicting sets of classes or sets/distributions of probabilities [20]: E.g., the set of all probabilities reflects total epistemic uncertainty, while a single probability (a one-point set) reflects no epistemic uncertainty.

## 3   Uncertainty and Fairness

**Prediction task**   The fairML literature has paid considerable attention to questions on whether and how to best include sensitive features in prediction modeling (e.g. in [35, 16]). Further, several error metrics have been constructed that reasonably measure some idea of fairness [4, 7, 29, 32]. We, in addition, argue that fair predictive models need to be uncertainty-aware models: When heteroscedasticity is not taken into account, a model cannot be fair, as predictions have different meanings across individuals. In order to predict in a fair manner, the model needs to account for its own ignorance. Dealing with **aleatoric** uncertainty, one example is to include the social reality of heteroscedasticity by using a distributional regression [37]. The main advantage is the interpretability property: Further parameters than the conditional mean, e.g. the conditional variance, can be modeled and corresponding feature importances can be calculated. Such measures of uncertainty can be used to investigate questions like: For which social groups is there more information in the data? Which features are collected and engineered in an informative manner? How big is the uncertainty for a given

individual? We note that although distributional regression offers good interpretability properties, it depends heavily on its distributional assumption. Conformal quantile regressions, in contrast, do not rely on such assumptions to lead to valid uncertainty quantifications but are less straight forward to be interpreted [10]. **Epistemic** uncertainty poses other questions relevant to fairness: Has enough data been collected for everyone? Are certain social groups underrepresented in the data?

Semi-structured deep distributional regression (SSDDR) as introduced in [38] combines structured additive distributional regression and neural networks. Hence, it provides scalability and flexibility through neural networks, direct interpretability through linear/smooth additive formula, and (aleatoric) uncertainty quantification through a distributional assumption [39]. All parameters $\theta_k, k = 1, \ldots, K$, of the distribution of $Y|(X, A)$, e.g., expectation and variance, can be modeled on their own. Typically in ADM systems, for certain features $A$, fairness considerations are particularly important, while other features $X$ can be used to optimize performance. This could be reflected in the specifications of SSDDR models:

$$\theta_k(X, A) = h_k(f_A(A) + f_X(X)) \tag{1}$$

Here, $f_A$ could be modeled in a structural additive fashion in order to ensure interpretability and control over the modeled effect structure. Regularization of the influence of $A$ may be used to comply with fairness metrics, when any indirect influence of $A$ over $X$ is also eliminated (e.g. by following [40]). $f_X$, in contrast, can be estimated in a more flexible manner, e.g. through a neural network, as fairness considerations may not directly apply. $h_k$ connects the features with the modeled parameter, e.g. $h_k(.) = \exp(.)$ ensures positivity for a variance parameter [38].

**Decision task** In order to create a fair decision system, a safe fail option needs to be incorporated [43]. We argue that this similarly applies to data-driven decision-making: If the model uncertainty is too high, any data-driven decision would violate the basic human rights of the individuals [33]. This aligns with fairness concerns that have been raised against the standard threshold decision function. Two individuals who are arbitrarily close to the threshold but on the other side of it are not treated in a similar way [3][p.96], which violates individual fairness [13]. One way to overcome this issue is to randomize decisions in a certain area where model uncertainty is high [3]. Another possibility is to define uncertainty regions and adjust classifications in the region using some fairness conceptions [22]. However, such solutions do not take the socio-technical reality of ADM systems into account: When data-driven decisions are associated with too high levels of uncertainty, decisions can still be made by other components of the system, e.g., human caseworker. Classification with reject option [19] offers the third option of non-data-driven classification: When the uncertainty is too high, instead of selecting any class, no class at all is selected in a data-driven fashion. Formally, when using the predicted probability of a certain event $P(Y = 1|X)$ to create a decision, a decision function with reject option can be written as [19]:

$$d_f(X) = \begin{cases} 0, & P(Y = 1|X) < \delta \\ 1, & P(Y = 1|X) > 1 - \delta \\ 2, & \text{else} \end{cases} \tag{2}$$

with $\delta \leq 0.5$ as a threshold[2] and decision 2 as reject option. This formulation can also be generalized for non-symmetric uncertainty regions, as in [19]. In an ADM system with a reject option individuals who are predicted to fall into the third category of "too uncertain to decide" [19] can be forwarded to the next instance of the system. This is especially useful when the data-driven decisions are considered cheaper but potentially less accurate, while the next decision instance, e.g. well-trained caseworkers (human experts), are considered more expensive but also more accurate.

## 4 Uncertainty aware profiling of job seekers

We demonstrate the use of SSDDR with reject option with an example of algorithmic profiling of job seekers. We use a large, anonymized sample of administrative labor market records provided by the German Institute for Labour Market and Employment Research (IAB, [1]), which enables us to model a realistic use case of public profiling. Decisions on the allocation of resources are made using either the predicted *duration of unemployment* ($T$, measured in months) or *long term unemployment* ($Y_{LTU}$), a binary version of the duration using 12 months as a threshold. Social injustice in the labor market

---

[2]$\delta = 0.5$ leads to the standard decision function.

has been documented regarding multiple dimensions, e.g. with respect to gender [6] and ethnicity [46]. Thus, in our example, $A$ includes gender (male/female) and citizenship (German/Non-German), while other features $X$ measure (un)employment histories, further socio-demographic characteristics, and other information captured in the administrative data (see Appendix A for details).

**Prediction task**  In order to investigate predictors of severe unemployment and uncertainty, the *duration of unemployment* was modeled using a two-parametric $\Gamma$ distribution (via SSDDR; referred to as GammaLIN). *Long term unemployment* was modeled using the one-parametric Bernoulli distribution (which equals to logistic regression; referred to as BinLIN). Since model performance is very similar for structured models and other ML approaches for the present task as suggested by [23], we chose the most simple model structure for the SSDDR models: both $f_A$ and $f_X$ were assumed to be additive and linear for all outcome parameters ($\mu, \sigma^2$) with an $L_1$-penalization for the coefficients of $f_X$ (see also Appendix A). In this way, we can achieve direct interpretability and compare feature importances, as all features were standardized between 0 and 1.

Table 1 presents the coefficients for both expectation and variance of GammaLIN for the four levels of $A$, given all other features are set to 0. Note that in this parametrization, expectation and variance are not independent but $\sigma_f = \frac{\mu^2}{\sigma^2}$.

| Social group A | Expectation $\mu$ | Variance $\sigma^2$ | Variance Factor $\sigma_f$ |
|---|---|---|---|
| Male - German | 0.87 | 0.76 | 1.00 |
| Female - German | 1.11 | 1.24 | 1.01 |
| Male - Non-German | 1.42 | 2.74 | 1.36 |
| Female - Non-German | 1.50 | 2.99 | 1.33 |

Table 1: Coefficients of the social groups based on SSDDR models for the duration of unemployment (GammaLIN).

While no strong differences can be observed between the male and female categories, being a Non-German citizen not only increases the *duration of unemployment* in expectation, but also in variance. Hence, for Non-German citizens, longer unemployment durations are predicted, but the model is also more uncertain about these predictions. However, note that effects on expectation and variance do not necessarily need to be in line. Table 3 in the Appendix shows the top 5 predictors for the expectation and variance of unemployment duration. Only one feature (the duration job seekers previously received unemployment benefits) is included in both lists of the most important predictors.

**Decision task**  In our example, interventions such as support programs are commonly assigned based on the prediction of long-term unemployment ($Y_{LTU}$, see [11]). To classify job seekers, we set up a decision function with a safe fail option. Different values of $\delta$ are used in the decision function, indicating different proportions of individuals who are actually classified either as long-term unemployed or non-long-term unemployed:

$$d_f(X) = \begin{cases} 0, & P(Y_{LTU} = 1|X, A) < \delta \\ 1, & P(Y_{LTU} = 1|X, A) > 1 - \delta \\ 2, & \text{else} \end{cases}$$

When modeling the *duration of unemployment*, $P(Y_{LTU} = 1|X, A)$ is calculated as $P(Y_{LTU} = 1|X, A) = 1 - F(T < 12|X, A)$, where $T$ denotes the duration in months.

In Figure 1, we plot classification accuracy against the proportion of individuals who are not in the reject option and therefore part of the automated decision process. Both overall accuracy and accuracy per social group are shown. The overall accuracy curve resembles the curves for German (fe)males, while for Non-Germans both the the minimal accuracy and the shape of the curves differ: For Non-German females the models achieve lowest accuracy for full data-driven decisions; decreasing the proportion of data-driven classification, however, leads to some convergence of accuracy across groups. Figure 2 in the Appendix shows that Non-Germans males and females fall into the reject option at different rates as the proportion of data-driven decisions decreases.

Clearly, the fewer people are actually classified, the higher the overall accuracy. Notably, both models struggle to detect individuals who in fact become long-term unemployed. Instead, they work quite

well in detecting non-long-term unemployment. In this case, classification with reject option can help to detect individuals who are more likely to be misclassified (and not clearly at low risk of long-term unemployment). These individuals could be transferred to experienced caseworkers in employment agencies to determine the final decision. In the end, both ethical and economic criteria need to be considered when setting the threshold for manual versus automated decision making.

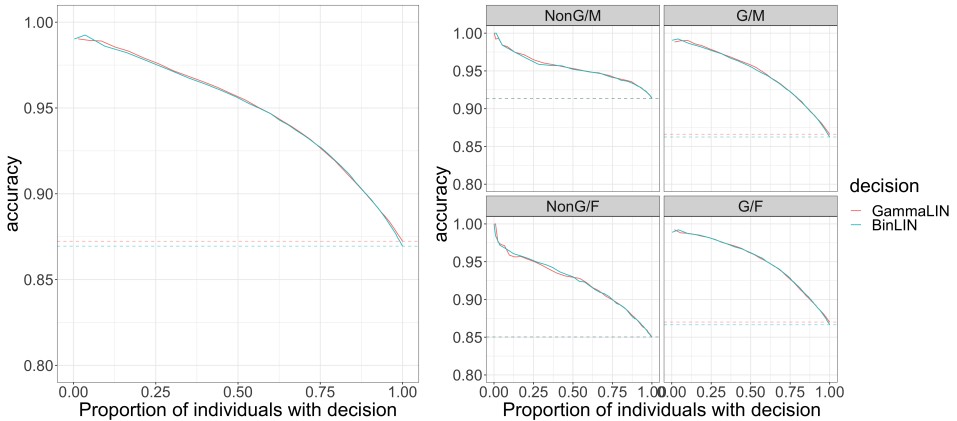

Figure 1: Accuracy of the Gamma (GammaLIN) and Bernoulli (BinLIN) models given different values of $\delta$. The x-axis shows the proportion of individuals who are not in the reject option, the dotted lines indicate baseline accuracy without a reject option. Left panel: Overall relationship; Right panel: Relationship by social group (NonG/M: Non-German Male, G/M: German Male, NonG/F: Non-German Female, G/F: German Female).

## 5 Discussion

In this paper, we argued that considering uncertainty is essential to build fair data-driven decision systems. We put the focus on the (aleatoric) uncertainty inherent in the data and investigated fairness implications at both the prediction and decision step. For the prediction task, uncertainty-aware models are needed to acknowledge all information that is included in the data about the social reality of individuals. For the decision task, we argue that the probabilistic information of the model output needs to be taken seriously. Therefore, ADM systems could include a safe fail option for cases where the ambiguity is too big.

We proposed SSDDR as a modeling framework for uncertainty-aware ADM. It allows to build flexible models that are interpretable with respect to a predefined set of (sensitive) features. SSDDR can account for both fairness considerations using a semi-structured predictor and uncertainty considerations using a distributional regression approach. We exemplified the use of SSDDR in a setting that closely mimicked a real-world application of algorithmic profiling with administrative data. The application highlighted that we can learn more about the social reality reflected in the data by considering the conditional variance of the distribution. Also, we proposed that an uncertainty-aware system can combine caseworker and data-driven decisions by using a reject option.

Further, SSDDRs flexibility would permit including other data, e.g. unstructured data (images, text) in the model using a suitable network architecture. This could, e.g., allow the inclusion of caseworker knowledge in the decision process by accounting for unstructured text data from interviews. Ideally, this could also lead to more fair data-driven systems assuming the case workers texts are fair.

As costs between decisions often vary, the proposed symmetric reject option might be limiting in practice. However, as in [19], classification with reject option can be generalized to situations with unequal costs using only one additional parameter. It results in a non-symmetric reject option which is not centered around the maximum level of aleatoric uncertainty (0.5). This would give developers and stakeholders even more control about the allocation of resources and the implicit distributive justice principles. As costs can vary across the sensitive attributes $A$, the decision function can be further customized using separate functions for different social groups.

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

| | Unemployment episodes | LTU episodes | Individuals | Individuals experiencing at least one LTU episode |
|---|---|---|---|---|
| **2015** (Train) | 86,692 | 12,688 (14.6%) | 76,187 | 12,688 (16.7%) |
| **2016** (Test) | 89,710 | 11,508 (12,8%) | 78,373 | 11,508 (14.7%) |

Table 2: LTU episodes and affected individuals

# A   Appendix

**Administrative data**   We use a 2% random sample of German administrative labor market records, called *Sample of Integrated Employment Biographies* (SIAB, [1]). The data combine information from various sources such as employment information, unemployment information and unemployment benefits receipt. Specifically, we use the factually anonymous version of the SIAB (SIAB-Regionalfile) – Version 7517 v1. Data access was provided via a Scientific Use File supplied by the Research Data Centre (FDZ) of the German Federal Employment Agency (BA) at the Institute for Employment Research (IAB). The observation level of the data is the unemployment episode. We restrict the SIAB data to unemployment episodes that occurred during the year 2015 for model training and used episodes from 2016 as the test set. The task is to predict the risk of long term unemployment (LTU) at the onset of a new unemployment episode, using information about the individuals' labor market history, the last job held, and socio-demographic characteristics (153 predictors in total, described in [23]).

**Model specification**   For both the Gamma and Bernoulli model, an equivalent model specification was used. No deep networks are included in the SSDDR models. The social groups ($A \in \mathbb{R}^4$) are included as unpenalized linear components, age ($X_{age} \in \mathbb{R}$) using a penalized B-Spline. The other features ($X_{other} \in \mathbb{R}^{153}$) are included linearly with $L_1$-regularization. The additive predictor of the model is therefore given as

$$\theta_k = h_k(A\beta_0^4 + f(X_{age}) + X_{other}\beta_1^{153}),$$

with $\beta_0^4 \in \mathbb{R}^4$, $\beta_1^{153} \in \mathbb{R}^{153}$, $h_k(.) = \exp(.)$ for $\mu$ and $\sigma^2$ of the Gamma model and $h_k(.) = \frac{\exp(.)}{1+\exp(.)}$ for the Bernoulli model.

The regularization parameter was tuned for the train year 2015 (test year 2016): First, $\lambda$ was searched on a very coarse grid to determine a first search interval, and afterward tuned on a grid $\lambda \in \{0.0001, \ldots, 0.05\}$ on the logarithmic scale. $\lambda = 2.6 \times 10^{-5}$ was found to be optimal. 60 epochs were used to optimize and train the models.

**Additional results**

| Expectation | | Variance | |
|---|---|---|---|
| **Name** | **Factor** | **Name** | **Factor** |
| LHG total | 14.4 | LEH total | 0.3 |
| seeking tot dur by age | 3.7 | LHG total | 1.6 |
| emp total dur | 2.6 | almp aw total | 1.5 |
| tsince lm contact | 2.4 | industry tot dur | 0.6 |
| emp total dur by age | 0.5 | est total | 1.5 |

Table 3: Top 5 most important predictors for expectation and variance of unemployment duration (GammaLIN). LHG total: Duration receiving unemployment benefits (ALG2); seeking tot dur by age: Duration of job seeking episodes by age; emp total dur: Duration in employment; tsince lm contact: Days since last labour market contact; emp total dur by age: Duration in employment by age; LEH total: Duration receiving unemployment benefits (ALG1); almp aw total: Number of participation in active labour market programs; industry tot dur: Duration worked in industry; est total: Number of different establishments worked in.

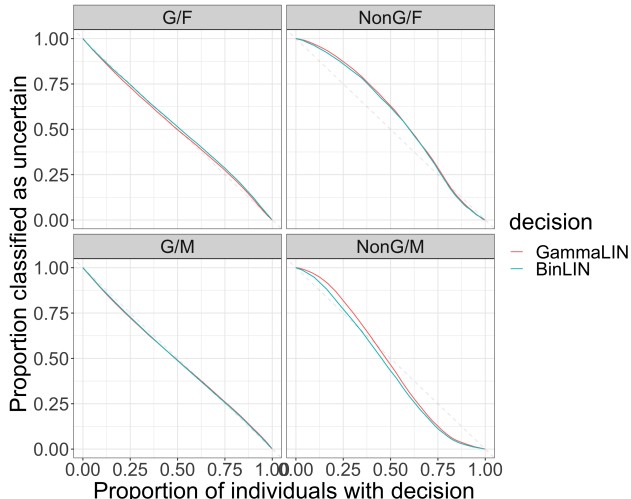

Figure 2: Proportion of individuals in the reject option of the gamma (GammaLIN) and bernoulli (BinLIN) models given different values of $\theta$. The x-axis shows the proportion of individuals who are not in the reject option. Results shown by social group (NonG/M: Non-German Male, G/M: German Male, NonG/F: Non-German Female, G/F: German Female).

