# OpenReview forum: "Uncertainty-aware predictive modeling for fair data-driven decisions"
_NeurIPS.cc/2022/Workshop/TSRML — TSRML2022_

### Official Review · Reviewer_yrcA · 2022-10-12
**An interesting but with limited contribution paper**

**Overall Rating:** 5

**Summary:**

This paper connects uncertainty with fairness in automated decision-making systems. By using a profiling task, the authors described how uncertainty can support more safe employment prediction. More specifically, with a safe fail option in ADM, the uncertain predictions were rejected from the automatic decision, and then the decision accuracy has been improved for the remaining data.




**Strengths:**

1. Overall, the paper is easy-to-follow, with a clear highlight on the importance of uncertainty for ADM.
2. It is also a novel study to involve uncertainty in the profiling of job seekers. The observations are insightful.



**Weaknesses:**

Despite the good flow of the paper, I still have some concerns:
1. The concept of uncertainty-aware decision-making is not new. Some references are missed, e.g.,
Bhatt, Umang, et al. "Uncertainty as a form of transparency: Measuring, communicating, and using uncertainty." Proceedings of the 2021 AAAI/ACM Conference on AI, Ethics, and Society. 2021.
2. This paper indeed failed to propose any new uncertainty quantification method that is designed for the decision task. The classification with the reject option is adapted from [18].
3. Some claims are quite confusing and need to be justified:
59  " In non-data driven decision-systems, on the other hand, there is no prediction task: the decision task is solved solely by human. experts or by predefined rules."
l150 "However, such solutions do not take the socio-technical reality of ADM systems into account: When data-driven decision systems are too uncertain to make a decision, decisions can still be made by other components of the system, e.g., human caseworkers. Classification with reject option [18] offers the third option of non-data-driven classification: When the uncertainty is too high, instead of selecting any class, the system selects no class at all"
Please stick to data-driven decision-making and clarify which method is applicable.





**Overall Recommendation:**

The topic of this paper is a good match with the scope of the workshop. Due to the limited technical contribution and unsurprising experimental results, it is hard to convince me of a strong acceptance.

**Review Confidence:**

3: The reviewer is fairly confident that the evaluation is correct

---

### Official Review · Reviewer_8DHK · 2022-10-18

**Overall Rating:** 6

**Summary:**

This paper intends to develop a fair decision-making system leveraging a distributional prediction model and a distribution-aware decision-making module. Specifically, the authors adopted a model that predicts the parameter of probability distributions. Then the authors used a decision-making policy that takes action only with confident prediction results. The model is evaluated on a labor market dataset.

**Strengths:**

Overall, this paper is well-written.
The motivation is quite good, and the proposed idea is easy to follow.

**Weaknesses:**

-The proposed method is quite straightforward as it directly combines a distributional regression module and a piecewise decision-making part.
-Given a supervised learning task, how to decide whether a feature is a sensitive attribute $A$ or feature $X$?
- Is it possible to evaluate the proposed model in terms of the fairness metrics such as demographic disparity?



**Overall Recommendation:**

Although the proposed method is not too sophisticated, I still believe this is a good work that initiates the study of fair decision-making systems.

**Review Confidence:**

4: The reviewer is confident but not absolutely certain that the evaluation is correct

---

### Official Review · Reviewer_o8Ab · 2022-10-20
**Interesting work with very nice presentation!**

**Overall Rating:** 9

**Summary:**

The authors highlight the importance of accounting uncertainty in automated decision-making (ADM) systems in order to further promote fairness and propose the use of the reject option in  ADM, which is triggered when the level of uncertainty is above a certain threshold. The authors use semi-structured deep distributional regression (SSDDR) to consider fairness-related features in addition to other predictive-informative features. The experiment results about job seekers profiling show the effectiveness of the safe fail option to omit to make predictions for those who are more likely to be classified, thus allowing handling of the decisions over to caseworkers for more humanely inquisition.

**Strengths:**

- The problem is well-motivated, and the proposed idea is presented very clearly
- The proposed approach seems solid, combining two well-known works (SSDDR and classification with reject option)
- The experiment results showcase the appealing potential of the proposed idea

**Weaknesses:**

- No major weaknesses.

- The authors could add more information about how to practically train the proposed model, especially since it is known that using the reject option induces excess risk, thus making it harder to obtain a highly performant classifier. How to ensure that the classifier does not overly invoke the reject option?

- The use cases of the proposed idea seem to apply to problems where prediction and decision tasks can be decoupled. Would the concept be extendable to situations where the decision is the direct output given a set of features (consider e.g., reinforcement learning-type task)?

**Overall Recommendation:**

The paper is well written. The problem is highly relevant to the workshop topic. The proposed idea seems interesting and appears technically correct.

**Review Confidence:**

3: The reviewer is fairly confident that the evaluation is correct

---

### Decision · Program_Chairs · 2022-10-23

**Decision:**

Accept

**Comment:**

Following the majority of recommendations from reviewers, the submission is accepted.